# Exploring the factors influencing alarm fatigue in intensive care units nurses: A cross-sectional study based on latent profile analysis

Liya Li[1◉], Wenxia Zhang[2◉], Yanling Chen[3]*

1 School of Nursing, Inner Mongolia Medical University, Hohhot, Inner Mongolia, China, 2 Cardiovascular Medicine Department, Affiliated Hospital of Inner Mongolia Medical University, Hohhot, Inner Mongolia, China, 3 Medical Development Department, Affiliated Hospital of Inner Mongolia Medical University, Hohhot, Inner Mongolia, China

◉ These authors contributed equally to this work.
* 782659545@qq.com

## Abstract

### Objective

To identify potential categories of alarm fatigue among ICU nurses and to explore the differences in characteristics and influencing factors among different categories.

### Methods

Using convenience sampling, 597 ICU nurses from 12 tertiary public hospitals across 8 cities in the Inner Mongolia Autonomous Region of China were recruited from September 2024 to December 2024. A cross-sectional survey was conducted using the General Information Questionnaire, ICU Nurses' Alarm Fatigue Scale, Stanford Presenteeism Scale: Health Status and Employee Productivity, and Nurses' Emotional Labor Scale. Potential profiles of nurse alarm fatigue were analyzed, and the influencing factors of different profiles were explored by univariate analysis and multivariate logistic regression analysis.

### Results

The median alarm fatigue scale score was 26(IQR = 19.75–31), and the alarm fatigue of ICU nurses could be categorized into low fatigue-robust tolerance group (30.8%), moderate fatigue (54.4%), and high fatigue-negative coping group (14.9%). The regression analyses showed that the number of children, the frequency of night shifts, the health status and employee productivity score, and the emotional labor score were the main factors of the ICU factors influencing different potential categories of nurse alarm fatigue (P < 0.05).

**Data availability statement:** "All relevant data are within the paper and its Supporting Information files."

**Funding:** This project is funded by research projects of Higher Education Institutions in Inner Mongolia Autonomous Region (grant number:NJZY23144), awarded to YC. The funders had no role in study design, data collection and analysis, decision to publish, or preparation of the manuscript.

**Competing interests:** The authors have declared that no competing interests exist.

## Conclusion

ICU nurses alarm in Inner Mongolia exhibited moderate-to-high alarm fatigue with notable subgroup heterogeneity. Nursing managers should implement tailored interventions addressing profile-specific factors, such as workload adjustments and emotional support strategies, to mitigate alarm fatigue.

## Introduction

With the advancement of medical technology, monitoring capabilities in intensive care units (ICU) have been continuously enhanced. These monitoring systems can provide timely alerts regarding changes in the patient's physiological and critical information for clinical decision- making [1]. However, this may also result in a spike in alerts, many of which are false and invalid [2,3]. While these monitoring devices can significantly assist clinical nurses in taking immediate action, alarm overload has become a challenge in hospitals. Studies indicate that 85% to 99% of alerts are erroneous or insignificant [4,5]. Alarm fatigue refers to a decrease in sensitivity and responsiveness to alarms or even desensitization of healthcare professionals due to continuous exposure to excessive alarms [6]. This condition not only undermines the quality of care and increases patient safety risks but also threatens the physical and psychological health of healthcare professionals [7]. One study estimated that a single ICU can receive up to 987 cardiac monitoring alerts per day [3]. The average daily noise levels in such environments also significantly exceed the maximum thresholds recommended by the World Health Organization (WHO) [8]. The Joint Commission on Accreditation of Healthcare Organizations (JCAHO) Adverse Event Database reported 80 patient deaths as a result of 98 alarm fatigue-related adverse events between 2009 and 2012 [9].In actuality, the frequency of adverse events linked to alarm fatigue might be higher than previously reported. Alarm fatigue poses critical risks that threaten patient safety, care quality, nurses' occupational health, and healthcare resource utilization [10].

Current research on alarm fatigue among ICU nurses mainly focuses on the present situation of alarm fatigue, its impact on nurses' psychology, and the exploration of alarm management strategies [11–13]. Nevertheless, there is a relative lack of research on the latent characteristics and influencing factors of alarm fatigue among ICU nurses in tertiary public hospitals in Inner Mongolia, China. Variations in economic development, healthcare resources, and work patterns suggest that ICU nurses in Inner Mongolia may face unique alarm fatigue challenges. Alarm fatigue among ICU nurses is strongly correlated with personal traits, health status and employee productivity, and emotional labor [14,15]. Research indicates that emotional labor significantly associated with alarm fatigue in the work of ICU nurses [14]. And excessive consumption of psychological resources affects the nurses' ability to process alarm information. Furthermore, poor health status and employee productivity can also make it difficult for nurses to remain full attention to their tasks [15]. These cognitive delays in responding to alarm may further escalate pre-existing alarm fatigue.

Therefore, this study aims to assess the current situation of alarm fatigue among ICU nurses in tertiary public hospitals in Inner Mongolia Autonomous Region, China,

through a questionnaire survey. It is designed to identify different types or groups characteristics of alarm fatigue among ICU nurses in this region, so as to facilitate a deeper understanding of its complexity and diversity. Meanwhile, the findings can provide valuable insights for improving the working environment of ICU nurses and enhancing patient safety both regionally and nationwide.

## Materials and methods

### Study design and participants

A convenience sampling method was used to select 597 ICU nurses from 12 tertiary-level public hospitals in 8 cities in Inner Mongolia from September 14, 2024 to December 9, 2024 as survey participants. Sixty ICU nurses were conveniently selected in a tertiary-level hospital in Inner Mongolia in September 2024 for the pre-survey. The minimum sample size required for this study was calculated based on the sample size estimation formula for measures, $N = 4U_\alpha^2 S^2/\delta^2$ [16]. Based on the pre-survey data yielding a standard deviation S=7.635 and an allowable error $\delta = 1.43$, N = 223, and considering a 20% invalid sample, a sample of at least 267 samples was required. The inclusion of 597 clinical nurses exceeded the minimum sample size requirement, ensuring a robust dataset. Inclusion criteria of the study participants: (a) had obtained a certificate of nurse practice and registered; (b) working in ICU for ≥1 year; exclusion criteria: (a) those who were in training or on leave during the survey period; (b) internships and rotating and seconded nurses; (c) nurses who suffered from significant life disasters in the last six months.

### Ethical considerations

The objectives and significance of the research were communicated to participants through an online survey, fostering informed consent. The privacy and anonymity of all participants were rigorously protected, and the voluntary nature of participation was emphasized. Additionally, the confidentiality of all collected data was meticulously maintained. The study protocol was submitted to the ethics committee of the host hospital (Ethics Committee of Inner Mongolia Medical University Affiliated Hospital), and ethical review approval was obtained (KY2024092).

### Measurement instruments

**Demographic characteristics.** Demographic information was gathered using a survey tool designed by the researcher, informed by relevant literature and study requirements. The tool included 11 items such as gender, age, education, average monthly income, professional title, number of children, scheduling arrangements, and so on.

**ICU nurses' alarm fatigue questionnaire.** It was prepared by Iranian scholar Torabizadeh in 2016 to quantify alarm fatigue in ICU nurses [17],introduced into China in 2021 by scholar Liu Jie [18]. The scale consists of 13 items with four dimensions. Each entry was scored on a 5-point Likert scale from "always" to "never", with entries 1 and 9 scored in reverse, and the total score ranged from 0 to 52. Higher scores indicate more severe alarm fatigue among ICU nurses. The Cronbach's alpha of the Chinese scale was 0.771, and the re-test reliability was 0.966. In this study, Cronbach's alpha of the scale was 0.814, our results confirm the scale's robustness.

**Stanford presenteeism scale: health status and employee productivity.** Introduced and culturally adapted by Zhao Fang and other scholars in 2010 as the Chinese version of The Stanford Presenteeism Scale (SPS-6) [19,20]. The scale is primarily used to quantify an individual's impaired work productivity due to health problems. The scale consists of six items, categorized into two dimensions: work limitations and work energy. Each entry is rated on a 5-point Likert scale from "strongly disagree" to "strongly agree" on a scale of 1–5, with entries 5 and 6 being reverse scored. The total score of the scale ranges from 6 to 30, and the higher the score, the greater the loss of productivity due to working while sick. The Cronbach's alpha of the Chinese scale was 0.760. The Cronbach's alpha for this scale in our sample was 0.709.

**The scale of emotional labor for nurses.** The Scale of Emotional Labor for Nurses was compiled by Hong and Kim in 2018 [21], and translated and validated in Chinese by Ying Yao in 2021 [22]. The scale contains three dimensions

of specialized emotion regulation, patient-centered emotion suppression, and standardized emotion play, consisting of 16 specific entries. A Likert 5-point scale was used, with a total score of 16–80, with higher scores indicating a greater degree of emotional labor on the part of the nurse at work. The Cronbach's alpha of the Chinese scale was 0.862. The Cronbach's alpha for this scale in this study was 0.920.

## Data collection

The online questionnaire star platform (https://www.wjx.cn) was used as the data collection tool for this study. The questionnaire covers all the variables and entries required for the study. The purpose of the survey, its content and how to fill it out were fully described in the introductory section. It was also emphasized that submission of the questionnaire constituted consent to participate in the study. The nurses were assured that the questionnaire was anonymous, that there were no right or wrong answers, and that they could withdraw from the study at any time. The nursing departments of participating hospitals distributed the survey link or QR code through ICU head nurses to eligible participants. Nurses accessed the questionnaire page by clicking the link or scanning the QR code to complete and submit the questionnaire online. Screening questions were set at the beginning of the questionnaire for nurses to judge whether they met the inclusion criteria, and those who did not would not be able to continue to fill out the questionnaire. The IP address restriction function of the questionnaire platform ensured that each nurse could only fill in the questionnaire once. When analyzing the data, questionnaires with a response time of <2 min and those with the same option in the Alarm Fatigue Scale were screened out to ensure the authenticity and validity of the questionnaire.

## Statistical analysis

Descriptive statistical analysis of the general demographic characteristics of the nurses was conducted using SPSS 24.0 statistical software. We employed latent profile analysis (LPA) to identify distinct profiles of alarm fatigue among ICU nurses. As a person-centered clustering method, LPA classifies individuals into unobserved latent subgroups based on similar response patterns across observed continuous variables [23]. LPA was performed using Mplus 8.3 software and the scores of the dimensions of the Alarm Fatigue Scale were used as observational indicators. To determine the optimal number of latent profiles, several model fit indices were compared, including the Akaike Information Criterion (AIC), Bayesian Information Criterion (BIC), adjusted BIC (aBIC), Entropy, Lo-Mendell-Rubin test (LMR), and Bootstrapped Likelihood Ratio Test (BLRT). These indices collectively aid in evaluating the goodness of fit and the adequacy of the models, with lower AIC, BIC, and aBIC values, higher Entropy values closer to 1, and statistically significant LMR and BLRT results($P<0.05$) indicating a better model fit.

The statistical methods employed encompassed descriptive statistical calculations (means, standard deviations, percentages), chi-square ($x^2$) tests, rank-sum tests, and multinomial logistic regression analysis. The significance level was set at $\alpha=0.05$, with $P<0.05$ indicating statistically significant differences. For measurement data that do not follow a normal distribution, the median is used as a way to provide a statistical description. The Kruskal-Wallis $H$ test was used to compare differences in demographic characteristics across profile categories, addressing ordinal variables and non-normal distributions. Furthermore, to investigate the factors associated with the different latent profiles of alarm fatigue, multivariate Logistic Regression analysis was performed. Prior to regression modeling, multicollinearity diagnostics were performed. This analysis enabled us to examine the relationship between various predictors (e.g., sociodemographic factors, work-related factors) and the latent profile membership, providing insights into the factors that contribute to the variability in alarm fatigue experiences among ICU nurses.

## Results

### Participant characteristics

In this study, a survey was conducted among ICU nurses from 12 tertiary public hospitals in the Inner Mongolia Autonomous Region. After rigorous screening, 15 invalid questionnaires were excluded, resulting in the collection of 582 valid

questionnaires, yielding an effective recovery rate of 97.5%. Among the participants, 240 (41.2%) hailed from Hohhot City, 63 (10.8%) from Baotou City, 31 (5.3%) from Ordos City, 46 (7.9%) from Tongliao City, another 46 (7.9%) from Hulunbeier City, 67 (11.5%) from Bayannur City, 44 (7.5%) from Chifeng City, and 45 (7.7%) from Xing'anmeng City. Detailed general information regarding the remaining study participants is presented in Table 2.

## Potential profiling and naming of alarm fatigue in ICU nurses

In this study, four dimensions of the Alarm Fatigue Scale were used as observed variables to fit models 1–5, and the specific fitting indices of each model are shown in Table 1. Based on the analysis of these results, the Lo-Mendell-Rubin test (LMR) and Bootstrapped Likelihood Ratio test (BLRT) for Model 2, Model 3, and Model 4 were statistically significant ($P < 0.05$). In terms of model fitting performance, Model 3 demonstrated a significant reduction in Akaike Information Criterion (AIC), Bayesian Information Criterion (BIC), and adjusted BIC (aBIC) values compared to Model 1 and Model 2, indicating its superiority in fitting the data. Although Models 4 and 5 exhibited slightly lower AIC, BIC, and aBIC values, Model 3 was preferred due to the increased risk of overfitting associated with the complexity of higher-order models. Furthermore, Model 3 had an Entropy value of 0.841 (Entropy > 0.8), indicating good classification accuracy, and the probability of belonging to each latent class ranged from 0.911 to 0.954, both of which were superior to those of Model 4. Therefore, Model 3 was selected as the optimal model for this study.

Fig 1 illustrates the distribution of mean alarm fatigue severity scores across three latent profiles in ICU nurses. The x-axis represents four dimensions of the scale, while the y-axis represents the average score of each dimension. Based on the distribution characteristics of scale scores, the profiles were classified as: low fatigue-robust tolerance (30.8%), moderate fatigue (54.4%), and high fatigue-negative coping (14.9%). The alarm fatigue score of C1 category was Median = 16.5, IQR = 13–18, which was mild alarm fatigue, and the scores of all dimensions were lower than those of C2 category and C3 category, especially in the dimension of work performance which was obviously at a low value, so it was named as "low fatigue-robust tolerance group". The alarm fatigue score of C2 category is Median = 27, IQR = 24–30, and the scores of all dimensions are at a medium level, so it is named "moderate fatigue group". The alarm fatigue score of category C3 is Median = 39, IQR = 37–43, which is the highest level, especially the score of work performance dimension is significantly higher than that of categories C1 and C2, indicating that they lack effective coping strategies in the face of stress, and therefore it is named as "high fatigue-negative coping group".

## Difference in characteristics among the latent classes

According to the results of the study average monthly income, number of children, frequency of night shifts, frequency of overtime work, work satisfaction, health status and employee productivity score, and emotional labor score in the three categories of alarm fatigue were not the same, and the differences were statistically significant ($P < 0.05$), as shown in Table 2.

## 2.4 Multinomial logistic regression analysis

The 3 potential profiles of nurse alarm fatigue were used as dependent variables, and the variables with statistically significant differences in the univariate analysis were used as independent variables. Unsorted multicategorical logistic regression was used for analysis because the parallel test $P$ value was < 0.001. The results of the multivariate logistic regression analysis showed that the number of children, frequency of night shifts, health status and employee productivity, and emotional labor were the influencing factors of ICU nurses' alarm fatigue ($P < 0.05$), as shown in Table 3. Among these variables, participants without children were 83.6% less likely to be categorized in the"low fatigue-robust tolerance"group compared with participants with ≥2 children (B = −1.807, OR= 0.164, 95% CI: 0.055–0.494). In addition, nurses with a night shift frequency of 1–4 times per month were more likely to fall into the"high fatigue-negative coping"group (B = −1.237, OR= 0.290, 95% CI: 0.118–0.714). Furthermore, for each 1-point increase in the health status and employee productivity

**Table 1. ICU nurse alarm fatigue potential profile fit metrics (n = 582).**

| Model | AIC | BIC | aBIC | Entropy | P-value | | Class probability (%) |
|---|---|---|---|---|---|---|---|
| | | | | | LMRT | BLRT | |
| 1 | 11194.566 | 11229.498 | 11204.101 | – | – | – | – |
| 2 | 10744.173 | 10800.937 | 10759.667 | 0.903 | <0.001 | <0.001 | 0.825\0.175 |
| 3 | 10435.974 | 10514.571 | 10457.428 | 0.841 | <0.001 | <0.001 | 0.308\0.544\0.149 |
| 4 | 10387.557 | 10487.986 | 10414.970 | 0.792 | 0.001 | <0.001 | 0.109\0.463\0.284\0.145 |
| 5 | 10336.270 | 10458.532 | 10369.642 | 0.788 | 0.151 | <0.001 | 0.121\0.288\0.339\0.111\0.141 |

Note: AIC, Akaike Information Criterion; BIC, Bayesian Information Criterion; aBIC, adjusted Bayesian Information Criterion; LMRT, Lo–Mendell–Rubin test; BLRT, Bootstrap Likelihood Ratio test.

score, nurses showed 25.3% decreased odds of belonging to the"low fatigue-robust tolerance"group compared to the"high fatigue-negative coping"group (B = −0.292, OR= 0.747, 95% CI: 0.684–0.815). For each 1-point increase in emotional labor score, the"low fatigue-robust tolerance"group was less likely than the "high fatigue-negative coping"group (B = −0.131, OR= 0.877, 95% CI: 0.836–0.920). Multicollinearity was assessed via variance inflation factors (VIF) and tolerance statistics. All variables showed VIF < 5, confirming no substantial collinearity issues, as shown in S1 Table.

## Discussion

### Status of alarm fatigue and demographic characteristics of the different categories

The results of this study showed that the ICU nurses' alarm fatigue score was Median = 26, IQR = 19.75–31, consistent with the findings of Ming Yueh [24], LEWANDOWSKA K [25], and others. These results indicate that the alarm fatigue level among ICU nurses remains relatively high both domestically and internationally. This trend can be attributed to two factors: (i) outdated medical equipment in most ICU units and decentralized alarm equipment results in burdensome workflows and operational inefficiencies for nurses [26]; and (ii) frequent false and invalid alarms disrupt nurses' judgment, decrease work efficiency, and reduce their sensitivity to alarm signals [27].

To explore the heterogeneity among ICU nurses, LPA identified three distinct alarm fatigue profiles: (i) high fatigue-negative coping group; (ii) moderate fatigue group; (iii) low fatigue-robust tolerance group. LPA revealed three distinct alarm fatigue profiles (low, moderate, high), uncovering hidden heterogeneity that traditional methods might overlook. This approach supports more targeted interventions, which align with Leung's emphasis on precision in subgroup-specific strategies [28].

For the high fatigue-negative coping group: nurses in this group exhibited high alarm fatigue across all dimension scores. This group of nurses is mainly characterized by without children, night shifts 1–4 times per month, high level of health status and employee productivity, and high burden of emotional labor. Such nurses mostly face problems such as short working hours, insufficient clinical experience, and poor physical and mental health. And they have insufficient regulation of the balance between nursing work and their own orientation. When dealing with medical alerts, it is difficult for them to deal with it effectively due to their lack of ability and experience. Targeted measures such as training, optimizing the scheduling system, and providing psychological support reduce their fatigue.

With regard to the moderate fatigue group, as the largest proportion of the sample, nurses scored in the mid-range across all dimensions. Their alarm fatigue should be closely monitored to prevent escalation. Reasonable staffing arrangements and support could help maintain balance in their work and well-being.

For the low fatigue-robust tolerance group: Compared to the high fatigue-negative coping group, the moderate fatigue group had lower scores on all dimensions. With better physical and mental state, they managed alarm stress with greater ease. Their professionalism and experience could be used to mentor others and enhance team capacity through peer support and collaboration.

**Table 2. General information of ICU nurses and one-way analysis of 3 potential profiles of alarm fatigue.**

| Variables | Categories | Overall (*n*=582) | low fatigue-robust tolerance group (*n*=176) | Moderate fatigue group (*n*=319) | high fatigue-negative coping group (*n*=87) | Test statistics | P-value |
|---|---|---|---|---|---|---|---|
| Gender [*n*(%)] | female | 517 (88.8) | 159 (90.3) | 281 (88.1) | 77 (88.5) | $\chi^2$=0.591 | 0.744 |
| | male | 65 (11.2) | 17 (9.7) | 38 (11.9) | 10 (11.5) | | |
| Age [*n*(%)] | 18-25 | 42 (7.2) | 9 (5.1) | 21 (6.6) | 12 (13.8) | H=3.271 | 0.195 |
| | 26-30 | 146 (25.1) | 45 (25.6) | 81 (25.4) | 20 (23.0) | | |
| | 31-40 | 336 (57.7) | 101 (57.4) | 185 (58.0) | 50 (57.5) | | |
| | ≥41 | 58 (10) | 21 (11.9) | 32 (10.0) | 5 (5.7) | | |
| Education [*n*(%)] | blow bachelor's degree | 104 (17.9) | 36 (20.5) | 49 (15.4) | 19 (21.8) | H=4.056 | 0.132 |
| | bachelor's degree | 469 (80.6) | 140 (79.5) | 263 (82.4) | 66 (75.9) | | |
| | above bachelor's degree | 9 (1.5) | 0 (0.0) | 7 (2.2) | 2 (2.3) | | |
| Professional title [*n*(%)] | nurse | 113 (19.4) | 32 (18.2) | 61 (19.1) | 20 (23.0) | H=0.057 | 0.972 |
| | nurse practitioner | 194 (33.3) | 60 (34.1) | 115 (36.1) | 19 (21.8) | | |
| | charge nurse | 224 (38.5) | 73 (41.5) | 106 (33.2) | 45 (51.7) | | |
| | associate chief nurse and above | 51 (8.8) | 11 (6.3) | 37 (11.6) | 3 (3.4) | | |
| Position [*n*(%)] | nurse staff | 503 (86.4) | 153 (86.9) | 277 (86.8) | 73 (83.9) | $\chi^2$=2.762 | 0.598 |
| | nursing group leader | 64 (11.0) | 19 (10.8) | 32 (10.0) | 13 (14.9) | | |
| | head nurse and above | 15 (2.6) | 4 (2.3) | 10 (3.1) | 1 (1.1) | | |
| Average monthly income [*n*(%)] | <4000¥ | 109 (18.7) | 35 (19.9) | 50 (15.7) | 24 (27.6) | H=11.735 | 0.003 |
| | 4000-8000¥ | 349 (60.0) | 107 (60.8) | 189 (59.2) | 53 (60.9) | | |
| | >8000¥ | 124 (21.3) | 34 (19.3) | 80 (25.1) | 10 (11.5) | | |
| Number of children [*n*(%)] | 0 | 184 (31.6) | 49 (27.8) | 84 (26.3) | 51 (58.6) | H=27.953 | <0.001 |
| | 1 | 295 (50.7) | 98 (55.7) | 170 (53.3) | 27 (31.0) | | |
| | ≥2 | 103 (17.7) | 29 (16.5) | 65 (20.4) | 9 (10.3) | | |
| Night shift frequency [*n*(%)] | 0 | 81 (13.9) | 22 (12.5) | 49 (15.4) | 10 (11.5) | H=10.165 | 0.006 |
| | 1-4 per month | 84 (14.4) | 20 (11.4) | 38 (11.9) | 26 (29.9) | | |
| | 5-6 per month | 187 (32.1) | 47 (26.7) | 115 (36.1) | 25 (28.7) | | |
| | ≥7 per month | 230 (39.5) | 87 (49.4) | 117 (36.7) | 26 (29.9) | | |
| Frequency of over-time work [*n*(%)] | 0-1 per month | 306 (52.6) | 100 (56.8) | 173 (54.2) | 33 (37.9) | H=6.115 | 0.047 |
| | 2-3 per month | 143 (24.6) | 39 (22.2) | 72 (22.6) | 32 (36.8) | | |
| | ≥4 per month | 133 (22.9) | 37 (21.0) | 74 (23.2) | 22 (25.3) | | |
| Satisfaction [*n*(%)] | satisfied | 319 (54.8) | 121 (68.8) | 147 (46.1) | 51 (58.6) | H=24.682 | <0.001 |
| | neutral | 225 (38.7) | 49 (27.8) | 145 (45.5) | 31 (35.6) | | |
| | dissatisfied | 38 (6.5) | 6 (3.4) | 27 (8.5) | 5 (5.7) | | |
| Alarm fatigue score [Median (IQR)] | | 26 (19.75–31) | 16.5 (13–18) | 27 (24–30) | 39 (37–43) | H=453.95 | <0.001 |
| Health status and employee productivity score [Median (IQR)] | | 18 (14-20) | 14 (11–18) | 18 (16–20) | 20 (18–21) | H=106.375 | <0.001 |
| Emotional labor score [Median (IQR)] | | 64 (61–72) | 69 (63–74) | 63 (60–65) | 74 (71–80) | H=149.816 | <0.001 |

Median (IQR), Median (25th–75th percentile).

## Factors influencing alarm fatigue profile among ICU nurses

Logistic regression analysis showed that nurses without children tended to have higher levels of alarm fatigue. Previous studies have shown that nurses without children have more severe levels of emotional exhaustion than nurses with children [29]. Young and less experienced, these nurses may not have yet adapted to the high-intensity, fast-paced ICU

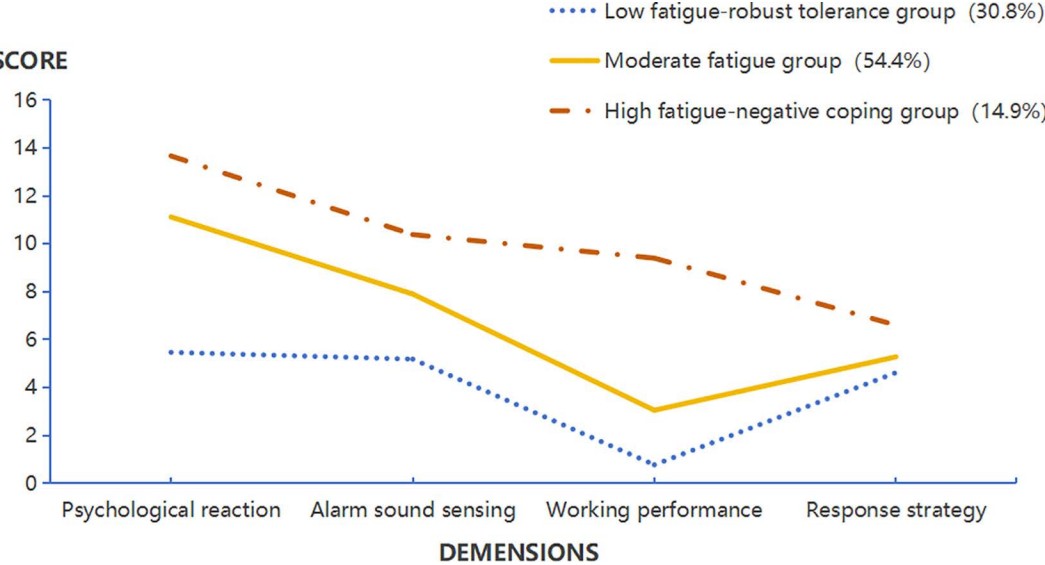

**Fig 1. Potential profiling of alarm fatigue in ICU nurses.**

environment. In contrast, nurses with children usually have caregiving experiences that enhance emotional regulation and stress resilience. These experiences make them more patient and resilient when dealing with complex situations and stress. Alarm management training, and support based on the Theory of Planned Behavior [30], may promote adaptive responses and alarm-related outcomes.

Alarm fatigue was also influenced by night shift frequency. Ming Yue et al. pointed out that night shifts were an independent influence on alarm fatigue [24]. And this study revealed that nurses working 1–4 night shifts per month experienced greater alarm fatigue compared to those working more than 4 night shifts monthly. This is possibly because of insufficient adaptability to irregular night work, which increases sensitivity to environmental stressors such as persistent alarms. Furthermore due to prolonged exposure to an environment filled with loud and frequent alarms. Nurses working high frequency night shifts may gradually become numb to such sounds and even subconsciously and selectively ignore them [31]. Just one study has shown that nurses working 12-hour shifts exhibit higher levels of alarm fatigue than nurses working 24-hour shifts [25]. Therefore, it is crucial to optimize shift scheduling and increase night shift exposure for less experienced staff, so as to facilitate their adaptation. It is also feasible to recalibrate alarm thresholds and reduce invalid alerts through evidence-based guidelines [32].

ICU nurses with higher scores on the Health Status and Employee Productivity Scale were more likely to be categorized in the high fatigue-negative coping group. That is, nurses with more severe health status and employee productivity had higher levels of alarm fatigue. Health status and employee productivity reflects nurses who are physically present at work but cannot fully perform due to health challenges [33]. ICU nurses often face greater physical and mental health challenges due to the high-intensity and high-stress nature of the work environment [34]. Chronic stress and insufficient recovery time can compromise cognitive processing and responsiveness. Cognitive load theory states that when task demands exceed processing capacity, it leads to cognitive overload, thus affecting the quality of performance and decision-making [35]. Therefore, managers can reduce extrinsic cognitive load by optimizing the design of alarm systems to reduce unnecessary alarms, clarifying alarm priorities and improving alarm presentation [36]. In addition, work should be done to optimize staff deployment and schedule enough rest periods to help preserve nurses' cognitive resources.

**Table 3. Results of multivariate logistic regression analysis (N = 582).**

| Variable | Low fatigue-robust tolerance group(ref. high fatigue-negative coping group) | | | | | | Moderate fatigue group(ref. high fatigue-negative coping group) | | | | | |
|---|---|---|---|---|---|---|---|---|---|---|---|---|
| | B | SE | OR | 95%CI | | P-value | B | SE | OR | 95%CI | | P-value |
| | | | | L | U | | | | | L | U | |
| Health status and employee productivity score | −0.292 | 0.045 | 0.747 | 0.684 | 0.815 | <0.001 | −0.094 | 0.042 | 0.910 | 0.838 | 0.989 | 0.026 |
| Emotional labor score | −0.131 | 0.025 | 0.877 | 0.836 | 0.920 | <0.001 | −0.217 | 0.024 | 0.805 | 0.767 | 0.844 | <0.001 |
| Average monthly income(ref. > 8000¥) | | | | | | | | | | | | |
| <4000¥ | 0.437 | 0.584 | 1.547 | 0.493 | 4.856 | 0.454 | −0.032 | 0.559 | 0.969 | 0.324 | 2.895 | 0.955 |
| 4000-8000¥ | −0.077 | 0.483 | 0.926 | 0.359 | 2.388 | 0.873 | −0.286 | 0.457 | 0.751 | 0.307 | 1.840 | 0.532 |
| Number of children(ref. ≥ 2) | | | | | | | | | | | | |
| 0 | −1.807 | 0.562 | 0.164 | 0.055 | 0.494 | 0.001 | −2.136 | 0.541 | 0.118 | 0.041 | 0.341 | <0.001 |
| 1 | −0.421 | 0.525 | 0.657 | 0.235 | 1.838 | 0.423 | −0.790 | 0.508 | 0.454 | 0.168 | 1.227 | 0.120 |
| Night shift frequency(ref. ≥ 7 per month) | | | | | | | | | | | | |
| 0 | −0.974 | 0.531 | 0.377 | 0.133 | 1.068 | 0.066 | −0.488 | 0.508 | 0.614 | 0.227 | 1.661 | 0.337 |
| 1-4 per month | −1.237 | 0.460 | 0.290 | 0.118 | 0.714 | 0.007 | −1.267 | 0.442 | 0.282 | 0.119 | 0.669 | 0.004 |
| 5-6 per month | −0.318 | 0.410 | 0.727 | 0.325 | 1.625 | 0.438 | 0.090 | 0.396 | 1.094 | 0.503 | 2.378 | 0.821 |
| Frequency of overtime work(ref. ≥ 4per month) | | | | | | | | | | | | |
| 0-1 per month | 0.467 | 0.420 | 1.595 | 0.701 | 3.632 | 0.266 | 0.556 | 0.400 | 1.743 | 0.796 | 3.815 | 0.165 |
| 2-3 per month | −0.110 | 0.452 | 0.896 | 0.369 | 2.171 | 0.807 | −0.337 | 0.429 | 0.714 | 0.308 | 1.656 | 0.432 |
| Satisfaction(ref. dissatisfied) | | | | | | | | | | | | |
| satisfied | −0.870 | 0.783 | 0.419 | 0.090 | 1.944 | 0.267 | −1.305 | 0.721 | 0.271 | 0.066 | 1.115 | 0.070 |
| neutral | −0.970 | 0.800 | 0.379 | 0.079 | 1.817 | 0.225 | −1.083 | 0.732 | 0.339 | 0.081 | 1.422 | 0.139 |

Note: SE, standard error; OR, odds ratio; B, unstandardized coefficient; CI, confidence interval; L: lower; U: upper.

Emotional labor was another significant predictor of alarm fatigue. Emotional labor is the process by which nurses need to manage and adjust their emotions to meet the demands of their work and the needs of their patients [37]. ICU is a high emotional labor load workplace [38]. They are obliged to remain composed and empathetic – often in the face of critical patient deterioration – which can deplete psychological reserves and reduce attention to alarms. A study by Ying Wang et al [14] has pointed out the correlation between emotional labor and workplace alarm fatigue. To help sustain emotional resilience and reduce alarm fatigue, it is important to set clear role definitions, minimize non-nursing tasks, and offer access to counseling or emotional support resources.

Notably, average monthly income showed no statistical significance (P > 0.05) in the multivariate logistic regression analysis. This phenomenon could be attributed to the relatively homogeneous monthly income levels among nurses in tertiary public hospitals within Inner Mongolia Autonomous Region, which constituted the primary sample source. Furthermore, work experience emerged as a significant confounding factor in the analysis, with experienced nurses demonstrating more efficient alert processing capabilities that potentially reduced alert fatigue levels. Potential interaction effects were observed between factors such as health status and employee productivity and night shift frequency. Shift work-induced physiological impairment may create synergistic effects by exacerbating both psychological stress and workload pressure [39], collectively contributing to elevated alert fatigue. Future studies should systematically investigate these confounding factors and interaction effects.

Beyond individualized interventions for nurses, systemic reforms are critical. Implementing machine learning algorithms for dynamic adjustment of alert thresholds and latency parameters could enhance system specificity. The integration of smartphones and wearable devices with intelligent filtering systems may facilitate prioritized alert recognition. National

health authorities should establish standardized alert management protocols referencing international guidelines like those from the American Heart Association [32] and American Association of Critical-Care Nurses [40], specifying setup criteria, response timelines, and documentation requirements. Multidisciplinary teams including clinicians, biomedical engineers, and informaticians should collaborate to develop comprehensive interventions addressing both technical and organizational dimensions of alarm fatigue.

## Limitations

First, the cross-sectional design of this study limited its ability to establish a causal relationship for the correlations examined in this study. Therefore, longitudinal studies can be used in subsequent research to explore the interrelationships between variables. Second, the study was confined to public tertiary hospitals in Inner Mongolia, China, limiting generalizability to other regions or lower-tier hospitals. The study sample was predominantly female, which limits generalizability, as more male participants may have produced different results. The female-majority sample limits broad applicability, as more male participants might yield different results. Third, a potential selection bias should be considered, given the convenience sampling approach. The use of an online questionnaire may have limited the participation of high-volume nurses due to the high intensity of ICU work, frequent shifts and unexpected tasks. The sample was skewed toward lower stress or time-appropriate groups. This may also result in biased results reflecting the true state of overall ICU nurses. Future studies should use stratified or random sampling to reduce selection bias and improve representativeness. Finally, all measurement outcomes were based on subjective reports, introducing social desirability and recall bias [41]. This may not be able to comprehensively and objectively reflect the alarm fatigue status of ICU nurses over a long period of time and under different workloads [42]. Future research should consider developing a more standardized and objective alarm fatigue measurement tool to advance the field.

## Conclusion

The alarm fatigue levels of ICU nurses can be categorized into three potential profiles: low fatigue-robust tolerance group, moderate fatigue group, and high fatigue-negative coping group. Nurses with different profiles had significant differences in the number of children, night shift frequency, health status and employee productivity, and emotional labor. Managers should take certain psychological support and interventions to reduce nurses' alarm fatigue as well as standardize the alarm management system and clarify the scope of responsibilities so as to ensure patient safety.

## Supporting information

**S1 Table. Multicollinearity Diagnostics for Independent Variables in the Regression Model.**
(DOCX)

**S2 Table. Raw data.**
(XLSX)

## Acknowledgments

The authors thank all the participants in this study and the nursing administrators at the collaborating hospitals for supporting the survey. All authors read and approved the final manuscript before submitting it to the journal for publication.

## Author contributions

**Data curation:** Liya Li.

**Formal analysis:** Yanling Chen.

**Investigation:** Liya Li.

**Methodology:** Liya Li.

**Resources:** Wenxia Zhang.

**Writing – original draft:** Liya Li.

**Writing – review & editing:** Wenxia Zhang, Yanling Chen.

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
