## [Decision Letter · Decision Letter 0]

Dear Dr. Chen,

Thank you for submitting your manuscript to PLOS ONE. After careful consideration, we feel that it has merit but does not fully meet PLOS ONE’s publication criteria as it currently stands. Therefore, we invite you to submit a revised version of the manuscript that addresses the points raised during the review process.

We look forward to receiving your revised manuscript.

Kind regards,

Othman A. Alfuqaha, Ph.D.

Academic Editor

PLOS ONE

Journal Requirements:

4. Please remove all personal information, ensure that the data shared are in accordance with participant consent, and re-upload a fully anonymized data set.

Additional Editor Comments :

Dear Authors,

Thank you for submitting your manuscript to PLOS ONE. We have received the reviewers' comments, and based on their feedback, my judgment is that your manuscript requires major revisions.

The reviewers have raised several concerns that need to be addressed to enhance the overall quality and clarity of your manuscript. I encourage you to carefully consider their comments and revise your manuscript accordingly.

Response to Major Concerns

1. Justification for using multiple psychological tools: Please do provide a clearer justification for including all instruments, addressing potential conceptual overlap and response burden.

2. Latent Profile Analysis (LPA) explanation: Please do add a brief description of LPA's purpose and its added value in understanding alarm fatigue patterns among ICU nurses.

3. Reliability statistics: Please do provide psychometric validation results, including Cronbach's alpha, to strengthen measurement rigor.

4. Sampling approach limitations: Please do elaborate on how the convenience sampling approach may affect external validity and interpretation of subgroup profiles.

5. Critical reflection and discussion: Please do deepen critical reflection, addressing unexpected or non-significant results, and discuss potential confounding factors or interaction effects.

6. Limitations section: Please do strengthen the limitations section by discussing potential biases introduced by the online self-administered survey method and the cross-sectional design.

Response to Minor Concerns

1. Latent profile naming: Please do clarify how the labels were derived and ensure consistency with underlying scale dimensions.

2. Discussion streamlining: Please do streamline the discussion section to avoid reiterating results without offering new insights.

3. Odds ratio interpretation: Please do add interpretation in plain language to improve accessibility.

4. Regression analysis diagnostic checks: Please do include details on diagnostic checks, such as multicollinearity and model fit.

5. Figure integration: Please do better explain how Figure 1 supports the interpretation of profiles.

6. Statistical term labeling: Please do ensure consistent formatting and labeling of statistical terms.

7. System-level strategies: Please do develop broader system-level strategies, such as alarm technology redesign and policy recommendations.

8. Language and grammar review: Please do conduct a minor language and grammar review to enhance readability.

Reviewers' comments:

Reviewer's Responses to Questions

**Comments to the Author**

1. Is the manuscript technically sound, and do the data support the conclusions?

Reviewer #1: Partly

Reviewer #2: Yes

2. Has the statistical analysis been performed appropriately and rigorously?

Reviewer #1: Yes

Reviewer #2: N/A

3. Have the authors made all data underlying the findings in their manuscript fully available?

Reviewer #1: Yes

Reviewer #2: Yes

4. Is the manuscript presented in an intelligible fashion and written in standard English?

Reviewer #1: Yes

Reviewer #2: No

Reviewer #1: Thank you for submitting this manuscript. The topic is timely and of interest to the general readership. Upon thorough evaluation, several major and minor concerns have been identified that should be addressed to enhance the overall quality and clarity of the manuscript.

Major concerns:

• The use of three different psychological tools (Alarm Fatigue Scale, Emotional Labor Scale, and Stanford Presenteeism Scale) may be excessive for a single cross-sectional study. While each tool is validated, the combination could introduce redundancy and participant fatigue. The authors should provide a clearer justification for including all instruments, especially considering potential conceptual overlap and increased response burden.

• The manuscript applies latent profile analysis (LPA) appropriately from a technical perspective, but it lacks an accessible explanation of the method for general readers. The authors should briefly describe LPA’s purpose and how it enhances the understanding of alarm fatigue patterns among ICU nurses. Furthermore, the discussion should better highlight the added value of using LPA over more traditional methods, such as how the profiles can guide targeted interventions.

• Although previously validated scales were used, the study does not report reliability statistics (e.g., Cronbach’s alpha) for the current sample. This raises concerns about the internal consistency and appropriateness of these tools in the specific cultural and professional context of ICU nurses in Inner Mongolia. The authors should provide psychometric validation results to strengthen the measurement rigor.

• The study uses convenience sampling across 12 hospitals, which may limit the representativeness of the findings. The authors acknowledge this limitation briefly, but they should elaborate on how the sampling approach may affect external validity and the interpretation of subgroup profiles, particularly in other healthcare settings.

• The discussion successfully addresses the major findings and situates them within the broader literature. However, it could benefit from deeper critical reflection, especially regarding any unexpected or non-significant results (e.g., income losing significance in the multivariate model). There is also limited attention to possible confounding factors or interaction effects, which could influence interpretation.

• The limitations are acknowledged, but the section could be strengthened by discussing potential biases introduced by the online self-administered survey method, including response bias and exclusion of less tech-savvy participants. In addition, the cross-sectional design limits causal inference, and this should be more clearly emphasized.

Minor concerns:

• The naming of latent profiles (e.g., “low fatigue–robust tolerance group” and “high fatigue–negative coping group”) may not be intuitive to readers. The authors should consider clarifying how these labels were derived and ensure consistency with the underlying scale dimensions.

• Portions of the discussion reiterate results already presented in the findings section without offering new insights. The authors may wish to streamline this section for clarity and impact.

• While odds ratios (ORs) are reported in the regression analysis, their clinical or practical significance is not explained. Adding interpretation in plain language (e.g., “nurses with X characteristic were Y times more likely to...”) would improve accessibility.

• The regression analysis lacks mention of diagnostic checks (e.g., multicollinearity, model fit). Including these details would enhance transparency and statistical rigor.

• Figure 1 is referenced but not well integrated into the narrative. The authors could better explain how this visual supports the interpretation of the profiles.

• In some instances, statistical terms like median and IQR are not clearly labeled or are inconsistently formatted (e.g., “26 (19.75, 31)” instead of “Median = 26, IQR = 19.75–31”).

• While the study addresses implications for nursing management, broader system-level strategies (e.g., alarm technology redesign, policy recommendations) are not fully developed.

• Some sentences, particularly in the abstract and introduction, are lengthy or awkwardly phrased. A minor language and grammar review would enhance readability.

Reviewer #2: I do think the authors had finished a instresting research ,and have some meanfuling results , these can give some advise to nuses for the effective alarm management. there are some suggestion for authors to revise,as follows and please visit the attachments.

My suggestions are, one is to check the statistical analysis and date,and another is to rivise the English langue ,smoothly.

**Do you want your identity to be public for this peer review?** For information about this choice, including consent withdrawal, please see our Privacy Policy

Reviewer #1: **Yes: ** mohd ismail ibrahim

Reviewer #2: No

---

## [Author Response · Author response to Decision Letter 1]

1 May 2025

Below is my concise response, and I have uploaded more information in the Word document responding to the reviewer.

Response to Editor:

We deeply appreciate your rigorous oversight and expeditious handling of our manuscript. I have responded to each of your questions below. We extend our heartfelt gratitude for your professional stewardship throughout the review process. The journal's exacting standards and your insightful feedback have profoundly strengthened this study's contribution to the field.

Response to Reviewer #1:

I would like to express my sincerest gratitude to you. Your review of our manuscript was meticulous, and your comments were professional and insightful, which deeply inspired me. Your feedback not only helped us identify the shortcomings in our article but also pointed out the direction for improvement, significantly enhancing the quality of our manuscript. It is my great honor to have such a conscientious and professional reviewer like you. Your feedback is a valuable learning opportunity for me, pushing me another step forward on the path of academic research. I look forward to contributing to the critical care nursing academic community by improving our research under your guidance. In addition, we will make further quality improvements with the ICU cardiac monitors to improve nurse alert fatigue and patient safety.

Below we provide point-by-point responses to your concerns, with all revisions highlighted in the "Revised Manuscript with Track Changes". In the responses below, I have italicized in blue the textual content after making changes in the manuscript for your review. Once again, thank you for your exceptional professionalism and dedication. Your guidance has been invaluable, and I am confident that the revised manuscript is significantly stronger thanks to your input.

Response to Reviewer #2:

I would like to extend my heartfelt thanks to you for your meticulous review and invaluable feedback on our manuscript. Your professional insights and constructive comments have greatly enhanced the quality of our work, and I am truly grateful for your time and effort. We have carefully reviewed and revised every point you raised, especially the language issues you highlighted. For instance, we have addressed the occasional digressions in narration and streamlined lengthy paragraphs to ensure focus and coherence. Your feedback on the discussion section was particularly enlightening. We have shortened sentences to ensure conciseness and impact. I feel incredibly fortunate to have benefited from your review, as it has highlighted areas for improvement that I would have otherwise overlooked. Your recognition of the study's significance also means a great deal to me, as it validates the importance of our research in addressing alarm fatigue among ICU nurses.

In response to your concern, we have conducted a comprehensive re-examination of all statistical procedures and data integrity. We are confident in the validity of our findings and welcome any further questions. Your scrutiny has strengthened the manuscript’s rigor, and we are grateful for your expertise.

In the responses below, I have italicized in blue the textual content after making changes in the manuscript for your review. I look forward to your continued feedback and the opportunity to further refine our work.

---

## [Decision Letter · Decision Letter 1]

plosone@plos.org

We look forward to receiving your revised manuscript.

Kind regards,

Othman A. Alfuqaha, Ph.D.

Academic Editor

PLOS ONE

Journal Requirements:

Additional Editor Comments:

Dear Authors,

Your paper has undergone significant improvements. However, I have a few minor comments to further enhance its quality.

Formatting

- Please review the PLoS ONE guidelines for font (12-point Times New Roman) and subheadings to ensure consistency throughout the manuscript.

Methods Section

- As your study adheres to the STROBE guidelines, please incorporate the recommended structure, including a "Study Design" subsection.

Discussion Section

- Consider removing subsections to improve the flow and coherence of the discussion.

References

- Verify that the references are formatted correctly in 12-point Times New Roman font.

Tables and Figures

- Add p-values where necessary in the tables.

- Enhance the clarity of the figure to improve readability.

By addressing these comments, you can further refine your manuscript.

Best regards,

Dr. Alfuqaha

Reviewers' comments:

Reviewer's Responses to Questions

**Comments to the Author**

Reviewer #1: All comments have been addressed

2. Is the manuscript technically sound, and do the data support the conclusions?

Reviewer #1: Yes

3. Has the statistical analysis been performed appropriately and rigorously?

Reviewer #1: Yes

4. Have the authors made all data underlying the findings in their manuscript fully available?

Reviewer #1: Yes

5. Is the manuscript presented in an intelligible fashion and written in standard English?

Reviewer #1: Yes

Reviewer #1: I would like to express my appreciation for the effort you have put into revising the manuscript. I am satisfied and genuinely pleased with the improvements made. Overall, the manuscript reads much better, demonstrates enhanced clarity and rigor, and reflects high quality work.

Well done, and congratulations on a job well executed. I am happy to recommend this version for publication.

**Do you want your identity to be public for this peer review?** For information about this choice, including consent withdrawal, please see our Privacy Policy

Reviewer #1: **Yes: ** Mohd Ismail

---

## [Author Response · Author response to Decision Letter 2]

23 May 2025

On behalf of all the authors, I would like to extend our heartfelt gratitude for your meticulous review and insightful feedback on our manuscript titled, "Exploring the factors influencing alarm fatigue in intensive care units nurses: A cross-sectional study based on latent profile analysis." We deeply appreciate your professionalism and efficiency throughout the review process.

After the last major revision, I learned a lot as well as recognized the shortcomings of the manuscript. We understand that a high-quality manuscript requires careful scrutiny and revision. We are truly grateful for your invaluable comments, which have not only helped us improve the manuscript but also broadened our research perspectives.

We have carefully addressed all your suggestions and made further improvements to ensure the manuscript meets the high standards of PLOS ONE. Should you have any further questions or suggestions regarding this revision, please do not hesitate to contact us. We would like to express our sincere gratitude once again for your hard work during the review process. We eagerly await your feedback and hope that our manuscript will be deemed suitable for publication in PLOS ONE, contributing to the research on alarm fatigue among ICU nurses.

---

## [Editor Report · Decision Letter 2]

Exploring the factors influencing alarm fatigue in intensive care units nurses: A cross-sectional study based on latent profile analysis

PONE-D-25-11224R2

Dear Dr.

<table border="0" cellpadding="0" cellspacing="0" class="datatable3" style="border-collapse: collapse; width: 678px; line-height: 14px; color: rgb(0, 0, 51); font-family: verdana, geneva, arial, helvetica, sans-serif; font-size: 11.2px;"> <tbody> <tr style="background-color: rgb(244, 244, 244);"> <td style="padding: 3px; border: 1px solid rgb(255, 255, 255);">Yanling Chen</td> </tr> <tr style="background-color: rgb(244, 244, 244);"> <td style="padding: 3px; border: 1px solid rgb(255, 255, 255); width: 196.094px;"> </td> </tr> </tbody></table>

,

We’re pleased to inform you that your manuscript has been judged scientifically suitable for publication and will be formally accepted for publication once it meets all outstanding technical requirements.

Kind regards,

Othman A. Alfuqaha, Ph.D.

Academic Editor

PLOS ONE

Additional Editor Comments (optional):

Congratulations! Your paper has been accepted with significant improvements.
---

## [Editor Report · Acceptance letter]

PONE-D-25-11224R2

PLOS ONE

Dear Dr. Chen,

I'm pleased to inform you that your manuscript has been deemed suitable for publication in PLOS ONE. Congratulations! Your manuscript is now being handed over to our production team.

Kind regards,

on behalf of

Dr. Othman A. Alfuqaha

Academic Editor

PLOS ONE